# Probabilistic Movement Primitives

**Alexandros Paraschos, Christian Daniel, Jan Peters, and Gerhard Neumann**
Intelligent Autonomous Systems, Technische Universität Darmstadt
Hochschulstr. 10, 64289 Darmstadt, Germany
{paraschos,daniel,peters,neumann}@ias.tu-darmstadt.de

## Abstract

Movement Primitives (MP) are a well-established approach for representing modular and re-usable robot movement generators. Many state-of-the-art robot learning successes are based MPs, due to their compact representation of the inherently continuous and high dimensional robot movements. A major goal in robot learning is to combine multiple MPs as building blocks in a modular control architecture to solve complex tasks. To this effect, a MP representation has to allow for blending between motions, adapting to altered task variables, and co-activating multiple MPs in parallel. We present a probabilistic formulation of the MP concept that maintains a distribution over trajectories. Our probabilistic approach allows for the derivation of new operations which are essential for implementing all aforementioned properties in one framework. In order to use such a trajectory distribution for robot movement control, we analytically derive a stochastic feedback controller which reproduces the given trajectory distribution. We evaluate and compare our approach to existing methods on several simulated as well as real robot scenarios.

## 1 Introduction

Movement Primitives (MPs) are commonly used for representing and learning basic movements in robotics, e.g., hitting and batting, grasping, etc. [1, 2, 3]. MP formulations are compact parameterizations of the robot's control policy. Modulating their parameters permits imitation and reinforcement learning as well as adapting to different scenarios. MPs have been used to solve many complex tasks, including 'Ball-in-the-Cup' [4], Ball-Throwing [5, 6], Pancake-Flipping [7] and Tetherball [8].

The aim of MPs is to allow for composing complex robot skills out of elemental movements with a modular control architecture. Hence, we require a MP architecture that supports parallel activation and smooth blending of MPs for composing complex movements of sequentially [9] and simultaneously [10] activated primitives. Moreover, adaptation to a new task or a new situation requires modulation of the MP to an altered desired target position, target velocity or via-points [3]. Additionally, the execution speed of the movement needs to be adjustable to change the speed of, for example, a ball-hitting movement. As we want to learn the movement from data, another crucial requirement is that the parameters of the MPs should be straightforward to learn from demonstrations as well as through trial and error for reinforcement learning approaches. Ideally, the same architecture is applicable for both stroke-based and periodic movements, and capable of representing optimal behavior in deterministic and stochastic environments.

While many of these properties are implemented by one or more existing MP architectures [1, 11, 10, 2, 12, 13, 14, 15], no approach exists which exhibits all of these properties in *one* framework. For example, [13] also offers a probabilistic interpretation of MPs by representing an MP as a learned graphical model. However, this approach heavily depends on the quality of the used planner and the

movement can not be temporally scaled. Rozo et. al. [12, 16] use a combination of primitives, yet, their control policy of the MP is based on heuristics and it is unclear how the combination of MPs affects the resulting movements.

In this paper, we introduce the concept of probabilistic movement primitives (ProMPs) as a general probabilistic framework for representing and learning MPs. Such a ProMP is a distribution over trajectories. Working with distributions enables us to formulate the described properties by operations from probability theory. For example, modulation of a movement to a novel target can be realized by conditioning on the desired target's positions or velocities. Similarly, consistent parallel activation of two elementary behaviors can be accomplished by a product of two independent trajectory probability distributions. Moreover, a trajectory distribution can also encode the variance of the movement, and, hence, a ProMP can often directly encode optimal behavior in stochastic systems [17]. Finally, a probabilistic framework allows us to model the covariance between trajectories of different degrees of freedom, that can be used to couple the joints of the robot.

Such properties of trajectory distributions have so far not been properly exploited for representing and learning MPs. The main reason for the absence of such an approach has been the difficulty of extracting a policy for controlling the robot from a trajectory distribution. We show how this step can be accomplished and derive a control policy that exactly reproduces a given trajectory distribution. To the best of our knowledge, we present the first principled MP approach that can exploit the power of operations from probability theory.

While the ProMPs' representation introduces many novel components, it incorporates many advantages from well-known previous movement primitive representations [18, 10], such as phase variables for timing of the movement that enable temporal rescaling of movements, and the ability to represent both rhythmic and stroke based movements. However, since ProMPs incorporate the variance of demonstrations, the increased flexibility and advantageous properties of the representation come at the price of requiring multiple demonstrations to learn the primitives as opposed to past approaches [18, 3] that can clone movements from a single demonstration.

## 2 Probabilistic Movement Primitives (ProMPs)

A movement primitive representation should exhibit several desirable properties, such as co-activation, adaptability and optimality in order to be a powerful MP representation. The goal of this paper is to unify these properties in one framework. We accomplish this objective by using a probabilistic formulation for MPs. We summarized all the properties and how they are implemented in our framework in Table 1. In this section, we will sequentially explain the importance of each of these property and discuss the implementation in our framework. As crucial part of our objective, we will introduce conditioning and a product of ProMPs as new

Table 1: Desirable properties and their implementation in the ProMP

| Property | Implementation |
|---|---|
| Co-Activation | Product |
| Modulation | Conditioning |
| Optimality | Encode variance |
| Coupling | Mean, Covariance |
| Learning | Max. Likelihood |
| Temporal Scaling | Modulate Phase |
| Rhythmic Movements | Periodic Basis |

operations that can be applied on the ProMPs due to the probabilistic formulation. Finally, we show how to derive a controller which follows a given trajectory distribution.

### 2.1 Probabilistic Trajectory Representation

We model a single movement execution as a trajectory $\boldsymbol{\tau} = \{q_t\}_{t=0...T}$, defined by the joint angles $q_t$ over time. In our framework, a MP describes multiple ways to execute a movement, which naturally leads to a probability distribution over trajectories.

**Encoding a Time-Varying Variance of Movements.** Our movement primitive representation models the time-varying variance of the trajectories to be able to capture multiple demonstrations with high-variability. Representing the variance information is crucial as it reflects the importance of

single time points for the movement execution and it is often a requirement for representing optimal behavior in stochastic systems [17].

We use a weight vector $\boldsymbol{w}$ to compactly represent a single trajectory. The probability of observing a trajectory $\boldsymbol{\tau}$ given the underlying weight vector $\boldsymbol{w}$ is given as a linear basis function model

$$\boldsymbol{y}_t = \left[ \begin{array}{c} q_t \\ \dot{q}_t \end{array} \right] = \boldsymbol{\Phi}_t^T \boldsymbol{w} + \boldsymbol{\epsilon}_y, \qquad\qquad p(\boldsymbol{\tau}|\boldsymbol{w}) = \prod_t \mathcal{N} \left( \boldsymbol{y}_t | \boldsymbol{\Phi}_t^T \boldsymbol{w}, \boldsymbol{\Sigma}_y \right), \qquad (1)$$

where $\boldsymbol{\Phi}_t = [\boldsymbol{\phi}_t, \dot{\boldsymbol{\phi}}_t]$ defines the $n \times 2$ dimensional time-dependent basis matrix for the joint positions $q_t$ and velocities $\dot{q}_t$, $n$ defines the number of basis functions and $\boldsymbol{\epsilon}_y \sim \mathcal{N}(\boldsymbol{0}, \boldsymbol{\Sigma}_y)$ is zero-mean i.i.d. Gaussian noise. By weighing the basis functions $\boldsymbol{\Psi}_t$ with the parameter vector $\boldsymbol{w}$, we can represent the mean of a trajectory.

In order to capture the variance of the trajectories, we introduce a distribution $p(\boldsymbol{w}; \boldsymbol{\theta})$ over the weight vector $\boldsymbol{w}$, with parameters $\boldsymbol{\theta}$. The trajectory distribution $p(\boldsymbol{\tau}; \boldsymbol{\theta})$ can now be computed by marginalizing out the weight vector $\boldsymbol{w}$, i.e., $p(\boldsymbol{\tau}; \boldsymbol{\theta}) = \int p(\boldsymbol{\tau}|\boldsymbol{w})p(\boldsymbol{w}; \boldsymbol{\theta})d\boldsymbol{w}$. The distribution $p(\boldsymbol{\tau}; \boldsymbol{\theta})$ defines a Hierarchical Bayesian Model (HBM) whose parameters are given by the observation noise variance $\boldsymbol{\Sigma}_y$ and the parameters $\boldsymbol{\theta}$ of $p(\boldsymbol{w}; \boldsymbol{\theta})$.

**Temporal Modulation.** Temporal modulation is needed for a faster or slower execution of the movement. We introduce a phase variable $z$ to decouple the movement from the time signal as for previous non-probabilistic approaches [18]. The phase can be any function monotonically increasing with time $z(t)$. By modifying the rate of the phase variable, we can modulate the speed of the movement. Without loss of generality, we define the phase as $z_0 = 0$ at the beginning of the movement and as $z_T = 1$ at the end. The basis functions $\boldsymbol{\phi}_t$ now directly depend on the phase instead of time, such that $\boldsymbol{\phi}_t = \boldsymbol{\phi}(z_t)$ and the corresponding derivative becomes $\dot{\boldsymbol{\phi}}_t = \boldsymbol{\phi}'(z_t)\dot{z}_t$.

**Rhythmic and Stroke-Based Movements.** The choice of the basis functions depends on the type of movement, which can be either rhythmic or stroke-based. For stroke-based movements, we use Gaussian basis functions $b_i^{\text{G}}$, while for rhythmic movements we use Von-Mises basis functions $b_i^{\text{VM}}$ to model periodicity in the phase variable $z$, i.e.,

$$b_i^{\text{G}}(z) = \exp\left( -\frac{(z_t - c_i)^2}{2h} \right), \quad b_i^{\text{VM}}(z) = \exp\left( \frac{\cos(2\pi(z_t - c_i))}{h} \right), \qquad (2)$$

where $h$ defines the width of the basis and $c_i$ the center for the $i$th basis function. We normalize the basis functions with $\phi_i(z_t) = b_i(z)/\sum_j b_j(z)$.

**Encoding Coupling between Joints.** So far, we have considered each degree of freedom to be modeled independently. However, for many tasks we have to coordinate the movement of the joints. A common way to implement such coordination is via the phase variable $z_t$ that couples the mean of the trajectory distribution [18]. Yet, it is often desirable to also encode higher-order moments of the coupling, such as the covariance of the joints at time point $t$. Hence, we extend our model to multiple dimensions. For each dimension $i$, we maintain a parameter vector $\boldsymbol{w}_i$, and we define the combined, weight vector $\boldsymbol{w}$ as $\boldsymbol{w} = [\boldsymbol{w}_1^T, \ldots, \boldsymbol{w}_n^T]^T$. The basis matrix $\boldsymbol{\Phi}_t$ now extends to a block-diagonal matrix containing the basis functions and their derivatives for each dimension. The observation vector $\boldsymbol{y}_t$ consists of the angles and velocities of all joints. The probability of an observation $\boldsymbol{y}$ at time $t$ is given by

$$p(\boldsymbol{y}_t|\boldsymbol{w}) = \mathcal{N}\left( \left[ \begin{array}{c} \boldsymbol{y}_{1,t} \\ \vdots \\ \boldsymbol{y}_{d,t} \end{array} \right] \Big| \left[ \begin{array}{ccc} \boldsymbol{\Phi}_t^T & \cdots & \boldsymbol{0} \\ \vdots & \ddots & \vdots \\ \boldsymbol{0} & \cdots & \boldsymbol{\Phi}_t^T \end{array} \right] \boldsymbol{w}, \boldsymbol{\Sigma}_y \right) = \mathcal{N}(\boldsymbol{y}_t|\boldsymbol{\Psi}_t\boldsymbol{w}, \boldsymbol{\Sigma}_y) \qquad (3)$$

where $\boldsymbol{y}_{i,t} = [q_{i,t}, \dot{q}_{i,t}]^T$ denotes the joint angle and velocity for the $i^{\text{th}}$ joint. We now maintain a distribution $p(\boldsymbol{w}; \boldsymbol{\theta})$ over the combined parameter vector $\boldsymbol{w}$. Using this distribution, we can also capture the covariance between joints.

**Learning from Demonstrations.** One crucial requirement of a MP representation is that the parameters of a single primitive are easy to acquire from demonstrations. To facilitate the estimation

of the parameters, we will assume a Gaussian distribution for $p(\boldsymbol{w}; \boldsymbol{\theta}) = \mathcal{N}(\boldsymbol{w}|\boldsymbol{\mu}_w, \boldsymbol{\Sigma}_w)$ over the parameters $\boldsymbol{w}$. Consequently, the distribution of the state $p(\boldsymbol{y}_t|\boldsymbol{\theta})$ for time step $t$ is given by

$$p(\boldsymbol{y}_t; \boldsymbol{\theta}) = \int \mathcal{N}\left(\boldsymbol{y}_t|\boldsymbol{\Psi}_t^T \boldsymbol{w}, \boldsymbol{\Sigma}_y\right) \mathcal{N}(\boldsymbol{w}|\boldsymbol{\mu_w}, \boldsymbol{\Sigma_w}) d\boldsymbol{w} = \mathcal{N}\left(\boldsymbol{y}_t|\boldsymbol{\Psi}_t^T \boldsymbol{\mu_w}, \boldsymbol{\Psi}_t^T \boldsymbol{\Sigma_w} \boldsymbol{\Psi}_t + \boldsymbol{\Sigma}_y\right), \quad (4)$$

and, thus, we can easily evaluate the mean and the variance for any time point $t$. As a ProMP represents multiple ways to execute an elemental movement, we also need multiple demonstrations to learn $p(\boldsymbol{w}; \boldsymbol{\theta})$. The parameters $\boldsymbol{\theta} = \{\boldsymbol{\mu_w}, \boldsymbol{\Sigma_w}\}$ can be learned from multiple demonstrations by maximum likelihood estimation, for example, by using the expectation maximization algorithm for HBMs with Gaussian distributions [19].

## 2.2 New Probabilistic Operators for Movement Primitives

The ProMPs allow for the formulation of new operators from probability theory, e.g., conditioning for modulating the trajectory and a product of distributions for co-activating MPs. We will now describe both operators in our general framework and, subsequently, discuss their implementation for our specific choice of Gaussian distributions for $p(\boldsymbol{w}; \boldsymbol{\theta})$.

**Modulation of Via-Points, Final Positions or Velocities by Conditioning.** The modulation of via-points and final positions are important properties of any MP framework such that the MP can be adapted to new situations. In our probabilistic formulation, such operations can be described by conditioning the MP to reach a certain state $\boldsymbol{y}_t^*$ at time $t$. Conditioning is performed by adding a desired observation $\boldsymbol{x_t} = [\boldsymbol{y}_t^*, \boldsymbol{\Sigma}_y^*]$ to our probabilistic model and applying Bayes theorem, i.e., $p(\boldsymbol{w}|\boldsymbol{x}_t^*) \propto \mathcal{N}\left(\boldsymbol{y}_t^*|\boldsymbol{\Psi}_t^T \boldsymbol{w}, \boldsymbol{\Sigma}_y^*\right) p(\boldsymbol{w})$. The state vector $\boldsymbol{y}_t^*$ represents the desired position and velocity vector at time $t$ and $\boldsymbol{\Sigma}_y^*$ describes the accuracy of the desired observation. We can also condition on any subset of $\boldsymbol{y}_t^*$. For example, by specifying a desired joint position $q_1$ for the first joint the trajectory distribution will automatically infer the most probable joint positions for the other joints.

For Gaussian trajectory distributions the conditional distribution $p(\boldsymbol{w}|\boldsymbol{x}_t^*)$ for $\boldsymbol{w}$ is Gaussian with mean and variance

$$\boldsymbol{\mu_w}^{[\text{new}]} = \boldsymbol{\mu_w} + \boldsymbol{\Sigma_w} \boldsymbol{\Psi}_t \left(\boldsymbol{\Sigma}_y^* + \boldsymbol{\Psi}_t^T \boldsymbol{\Sigma_w} \boldsymbol{\Psi}_t\right)^{-1} \left(\boldsymbol{y}_t^* - \boldsymbol{\Psi}_t^T \boldsymbol{\mu_w}\right), \quad (5)$$

$$\boldsymbol{\Sigma_w}^{[\text{new}]} = \boldsymbol{\Sigma_w} - \boldsymbol{\Sigma_w} \boldsymbol{\Psi}_t \left(\boldsymbol{\Sigma}_y^* + \boldsymbol{\Psi}_t^T \boldsymbol{\Sigma_w} \boldsymbol{\Psi}_t\right)^{-1} \boldsymbol{\Psi}_t^T \boldsymbol{\Sigma_w}. \quad (6)$$

Conditioning a ProMP to different target states is also illustrated in Figure 1(a). We can see that, despite the modulation of the ProMP by conditioning, the ProMP stays within the original distribution, and, hence, the modulation is also learned from the original demonstrations. Modulation strategies in current approaches such as the DMPs do not show this beneficial effect [18].

**Combination and Blending of Movement Primitives.** Another beneficial probabilistic operation is to continuously combine and blend different MPs into a single movement. Suppose that we maintain a set of $i$ different primitives that we want to combine. We can co-activate them by taking the products of distributions, i.e., $p_{\text{new}}(\boldsymbol{\tau}) \propto \prod_i p_i(\boldsymbol{\tau})^{\alpha^{[i]}}$ where the $\alpha^{[i]} \in [0, 1]$ factors denote the activation of the $i^{\text{th}}$ primitive. This product captures the overlapping region of the active MPs, i.e., the part of the trajectory space where all MPs have high probability mass.

However, we also want to be able to modulate the activations of the primitives, for example, to continuously blend the movement execution from one primitive to the next. Hence, we decompose the trajectory into single time steps and use time-varying activation functions $\alpha_t^{[i]}$, i.e.,

$$p^*(\boldsymbol{\tau}) \propto \prod_t \prod_i p_i(\boldsymbol{y}_t)^{\alpha_t^{[i]}}, \quad p_i(\boldsymbol{y}_t) = \int p_i(\boldsymbol{y}_t|\boldsymbol{w}^{[i]}) p_i(\boldsymbol{w}^{[i]}) d\boldsymbol{w}^{[i]}. \quad (7)$$

For Gaussian distributions $p_i(\boldsymbol{y}_t) = \mathcal{N}(\boldsymbol{y}_t|\boldsymbol{\mu}_t^{[i]}, \boldsymbol{\Sigma}_t^{[i]})$, the resulting distribution $p^*(\boldsymbol{y}_t)$ is again Gaussian with variance and mean

$$\boldsymbol{\Sigma}_t^* = \left(\sum_i \left(\boldsymbol{\Sigma}_t^{[i]}/\alpha_t^{[i]}\right)^{-1}\right)^{-1}, \quad \boldsymbol{\mu}_t^* = (\boldsymbol{\Sigma}_t^*)^{-1} \left(\sum_i \left(\boldsymbol{\Sigma}_t^{[i]}/\alpha_t^{[i]}\right)^{-1} \boldsymbol{\mu}_t^{[i]}\right) \quad (8)$$

Both terms, and their derivatives, are required to obtain the stochastic feedback controller which is finally used to control the robot. We illustrated the co-activation of two ProMPs in Figure 1(b) and the blending of two ProMPs in Figure 1(c).

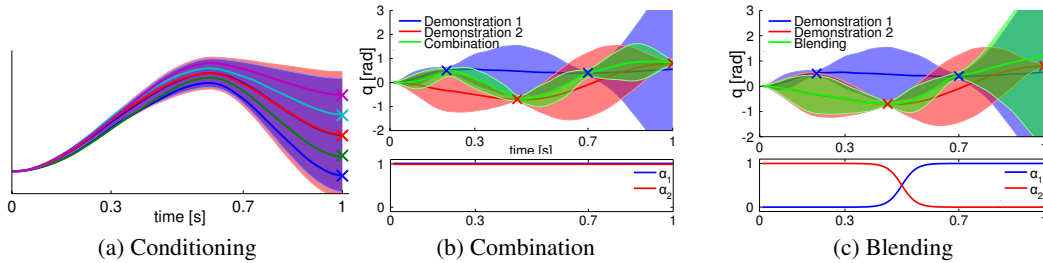

(a) Conditioning      (b) Combination      (c) Blending

Figure 1: (a) *Conditioning on different target states*. The blue shaded area represents the learned trajectory distribution. We condition on different target positions, indicated by the 'x'-markers. The produced trajectories exactly reach the desired targets while keeping the shape of the demonstrations. (b) *Combination of two ProMPs*. The trajectory distributions are indicated by the blue and red shaded areas. Both primitives have to reach via-points at different points in time, indicated by the 'x'-markers. We co-activate both primitives with the same activation factor. The trajectory distribution generated by the resulting feedback controller now goes through all four via-points. (c) *Blending of two ProMPs*. We smoothly blend from the red primitive to the blue primitive. The activation factors are shown in the bottom. The resulting movement (green) first follows the red primitive and, subsequently, switches to following the blue primitive.

## 2.3 Using Trajectory Distributions for Robot Control

In order to fully exploit the properties of trajectory distributions, a policy for controlling the robot is needed that reproduces these distributions. To this effect, we analytically derive a stochastic feedback controller that can accurately reproduce the mean vectors $\boldsymbol{\mu}_t$ and the variances $\boldsymbol{\Sigma}_t$ for all $t$ of a given trajectory distribution.

We follow a model-based approach. First, we approximate the continuous time dynamics of the system by a linearized discrete-time system with step duration $\mathrm{dt}$,

$$\boldsymbol{y}_{t+\mathrm{dt}} = \left(\boldsymbol{I} + \boldsymbol{A}_t\mathrm{dt}\right)\boldsymbol{y}_t + \boldsymbol{B}_t\mathrm{dt}\boldsymbol{u} + \boldsymbol{c}_t\mathrm{dt}, \tag{9}$$

where the system matrices $\boldsymbol{A}_t$, the input matrices $\boldsymbol{B}_t$ and the drift vectors $\boldsymbol{c}_t$ can be obtained by first order Taylor expansion of the dynamical system[1]. We assume a stochastic linear feedback controller with time varying feedback gains is generating the control actions, i.e.,

$$\boldsymbol{u} = \boldsymbol{K}_t\boldsymbol{y}_t + \boldsymbol{k}_t + \boldsymbol{\epsilon}_{\boldsymbol{u}}, \quad \boldsymbol{\epsilon} \sim \mathcal{N}\left(\boldsymbol{\epsilon}_{\boldsymbol{u}}|0, {}^{\boldsymbol{\Sigma}_u}\!/\!\mathrm{dt}\right), \tag{10}$$

where the matrix $\boldsymbol{K}_t$ denotes a feedback gain matrix and $\boldsymbol{k}_t$ a feed-forward component. We use a control noise which behaves like a Wiener process [21], and, hence, its variance grows linearly with the step duration[2] $\mathrm{dt}$. By substituting Eq. (10) into Eq. (9), we rewrite the next state of the system as

$$\boldsymbol{y}_{t+\mathrm{dt}} = \left(\boldsymbol{I} + \left(\boldsymbol{A}_t + \boldsymbol{B}_t\boldsymbol{K}_t\right)\mathrm{dt}\right)\boldsymbol{y}_t + \boldsymbol{B}_t\mathrm{dt}(\boldsymbol{k}_t + \boldsymbol{\epsilon}_u) + \boldsymbol{c}\mathrm{dt} = \boldsymbol{F}_t\boldsymbol{y}_t + \boldsymbol{f}_t + \boldsymbol{B}_t\mathrm{dt}\boldsymbol{\epsilon}_u,$$
$$\text{with } \boldsymbol{F}_t = \left(\boldsymbol{I} + \left(\boldsymbol{A}_t + \boldsymbol{B}_t\boldsymbol{K}_t\right)\mathrm{dt}\right), \quad \boldsymbol{f}_t = \boldsymbol{B}_t\boldsymbol{k}_t\mathrm{dt} + \boldsymbol{c}\mathrm{dt}. \tag{11}$$

For improved clarity, we will omit the time-index as subscript for most matrices in the remainder of the paper. From Eq. 4 we know that the distribution for our current state $\boldsymbol{y}_t$ is Gaussian with mean $\boldsymbol{\mu}_t = \boldsymbol{\Psi}_t^T\boldsymbol{\mu}_w$ and covariance[3] $\boldsymbol{\Sigma}_t = \boldsymbol{\Psi}_t^T\boldsymbol{\Sigma}_w\boldsymbol{\Psi}_t$. As the system dynamics are modeled by a Gaussian linear model, we can obtain the distribution of the next state $p\left(\boldsymbol{y}_{t+\mathrm{dt}}\right)$ analytically from the forward model

$$p\left(\boldsymbol{y}_{t+\mathrm{dt}}\right) = \int \mathcal{N}\left(\boldsymbol{y}_{t+\mathrm{dt}}|\boldsymbol{F}\boldsymbol{y}_t + \boldsymbol{f}, \boldsymbol{\Sigma}_s\mathrm{dt}\right)\mathcal{N}\left(\boldsymbol{y}_t|\boldsymbol{\mu}_t, \boldsymbol{\Sigma}_t\right)d\boldsymbol{y}_t$$
$$= \mathcal{N}\left(\boldsymbol{y}_{t+\mathrm{dt}}|\boldsymbol{F}\boldsymbol{\mu}_t + \boldsymbol{f}, \boldsymbol{F}\boldsymbol{\Sigma}_t\boldsymbol{F}^T + \boldsymbol{\Sigma}_s\mathrm{dt}\right), \tag{12}$$

where $dt\boldsymbol{\Sigma}_s = dt\boldsymbol{B}\boldsymbol{\Sigma}_u\boldsymbol{B}^T$ represents the system noise matrix. Both sides of Eq. 12 are Gaussian distributions, where the left-hand side can also be computed by our desired trajectory distribution $p(\boldsymbol{\tau};\boldsymbol{\theta})$. We match the mean and the variances of both sides with our control law, i.e.,

$$\boldsymbol{\mu}_{t+dt} = \boldsymbol{F}\boldsymbol{\mu}_t + (\boldsymbol{B}\boldsymbol{k} + \boldsymbol{c})dt, \qquad \boldsymbol{\Sigma}_{t+dt} = \boldsymbol{F}\boldsymbol{\Sigma}_t\boldsymbol{F}^T + \boldsymbol{\Sigma}_s dt, \tag{13}$$

where $\boldsymbol{F}$ is given in Eq. (11) and contains the time varying feedback gains $\boldsymbol{K}$. Using both constraints, we can now obtain the time dependend gains $\boldsymbol{K}$ and $\boldsymbol{k}$.

**Derivation of the Controller Gains.** By rearranging terms, the covariance constraint becomes

$$\boldsymbol{\Sigma}_{t+dt} - \boldsymbol{\Sigma}_t = \boldsymbol{\Sigma}_s dt + (\boldsymbol{A} + \boldsymbol{B}\boldsymbol{K})\boldsymbol{\Sigma}_t dt + \boldsymbol{\Sigma}_t (\boldsymbol{A} + \boldsymbol{B}\boldsymbol{K})^T dt + O(dt^2), \tag{14}$$

where $O(dt^2)$ denotes all second order terms in dt. After dividing by dt and taking the limit of $dt \to 0$, the second order terms disappear and we obtain the time derivative of the covariance

$$\dot{\boldsymbol{\Sigma}}_t = \lim_{dt\to 0} \frac{\boldsymbol{\Sigma}_{t+dt} - \boldsymbol{\Sigma}_t}{dt} = (\boldsymbol{A} + \boldsymbol{B}\boldsymbol{K})\boldsymbol{\Sigma}_t + \boldsymbol{\Sigma}_t(\boldsymbol{A} + \boldsymbol{B}\boldsymbol{K})^T + \boldsymbol{\Sigma}_s. \tag{15}$$

The matrix $\dot{\boldsymbol{\Sigma}}_t$ can also be obtained from the trajectory distribution $\dot{\boldsymbol{\Sigma}}_t = \dot{\boldsymbol{\Psi}}_t^T\boldsymbol{\Sigma}_w\boldsymbol{\Psi}_t + \boldsymbol{\Psi}_t^T\boldsymbol{\Sigma}_w\dot{\boldsymbol{\Psi}}_t$, which we substitute into Eq. (15). After rearranging terms, the equation reads

$$\boldsymbol{M} + \boldsymbol{M}^T = \boldsymbol{B}\boldsymbol{K}\boldsymbol{\Sigma}_t + (\boldsymbol{B}\boldsymbol{K}\boldsymbol{\Sigma}_t)^T, \text{ with } \boldsymbol{M} = \dot{\boldsymbol{\Phi}}_t\boldsymbol{\Sigma}_w\boldsymbol{\Phi}_t^T - \boldsymbol{A}\boldsymbol{\Sigma}_t - \boldsymbol{\Sigma}_s/2. \tag{16}$$

Setting $\boldsymbol{M} = \boldsymbol{B}\boldsymbol{K}\boldsymbol{\Sigma}_t$ and solving for the gain matrix $\boldsymbol{K}$

$$\boldsymbol{K} = \boldsymbol{B}^\dagger \left(\dot{\boldsymbol{\Psi}}_t^T\boldsymbol{\Sigma}_w\boldsymbol{\Psi}_t - \boldsymbol{A}\boldsymbol{\Sigma}_t - \boldsymbol{\Sigma}_s/2\right)\boldsymbol{\Sigma}_t^{-1}, \tag{17}$$

yields the solution, where $\boldsymbol{B}^\dagger$ denotes the pseudo-inverse of the control matrix $\boldsymbol{B}$.

**Derivation of the Feed-Forward Controls.** Similarly, we obtain the feed-forward control signal $\boldsymbol{k}$ by matching the mean of the trajectory distribution $\boldsymbol{\mu}_{t+dt}$ with the mean computed with the forward model. After rearranging terms, dividing by dt and taking the limit of $dt \to 0$, we arrive at the continuous time constraint for the vector $\boldsymbol{k}$,

$$\dot{\boldsymbol{\mu}}_t = (\boldsymbol{A} + \boldsymbol{B}\boldsymbol{K})\boldsymbol{\mu}_t + \boldsymbol{B}\boldsymbol{k} + \boldsymbol{c}. \tag{18}$$

We can again use the trajectory distribution $p(\boldsymbol{\tau};\boldsymbol{\theta})$ to obtain $\boldsymbol{\mu}_t = \boldsymbol{\Psi}_t\boldsymbol{\mu}_w$ and $\dot{\boldsymbol{\mu}}_t = \dot{\boldsymbol{\Psi}}_t\boldsymbol{\mu}_w$ and solve Eq. (18) for $\boldsymbol{k}$,

$$\boldsymbol{k} = \boldsymbol{B}^\dagger \left(\dot{\boldsymbol{\Psi}}_t\boldsymbol{\mu}_w - (\boldsymbol{A} + \boldsymbol{B}\boldsymbol{K})\boldsymbol{\Psi}_t\boldsymbol{\mu}_w - \boldsymbol{c}\right) \tag{19}$$

**Estimation of the Control Noise.** In order to match a trajectory distribution, we also need to match the control noise matrix $\boldsymbol{\Sigma}_u$ which has been applied to generate the distribution. We first compute the system noise covariance $\boldsymbol{\Sigma}_s = \boldsymbol{B}\boldsymbol{\Sigma}_u\boldsymbol{B}^T$ by examining the cross-correlation between time steps of the trajectory distribution. To do so, we compute the joint distribution $p\left(\boldsymbol{y}_t, \boldsymbol{y}_{t+dt}\right)$ of the current state $\boldsymbol{y}_t$ and the next state $\boldsymbol{y}_{t+dt}$,

$$p\left(\boldsymbol{y}_t, \boldsymbol{y}_{t+dt}\right) = \mathcal{N}\left(\left[\begin{array}{c} \boldsymbol{y}_t \\ \boldsymbol{y}_{t+dt} \end{array}\right] \middle| \left[\begin{array}{c} \boldsymbol{\mu}_t \\ \boldsymbol{\mu}_{t+dt} \end{array}\right], \left[\begin{array}{cc} \boldsymbol{\Sigma}_t & \boldsymbol{C}_t \\ \boldsymbol{C}_t^T & \boldsymbol{\Sigma}_{t+dt} \end{array}\right]\right), \tag{20}$$

where $\boldsymbol{C}_t = \boldsymbol{\Psi}_t\boldsymbol{\Sigma}_w\boldsymbol{\Psi}_{t+dt}^T$ is the cross-correlation. We can again use our model to match the cross correlation. The joint distribution for $\boldsymbol{y}_t$ and $\boldsymbol{y}_{t+dt}$ is obtained by our system dynamics by $p\left(\boldsymbol{y}_t, \boldsymbol{y}_{t+dt}\right) = \mathcal{N}\left(\boldsymbol{y}_t | \boldsymbol{\mu}_t, \boldsymbol{\Sigma}_t\right)\mathcal{N}\left(\boldsymbol{y}_{t+dt} | \boldsymbol{F}\boldsymbol{y}_t + \boldsymbol{f}, \boldsymbol{\Sigma}_u\right)$ which yields

$$p\left(\boldsymbol{y}_t, \boldsymbol{y}_{t+dt}\right) = \mathcal{N}\left(\left[\begin{array}{c} \boldsymbol{y}_t \\ \boldsymbol{y}_{t+dt} \end{array}\right] \middle| \left[\begin{array}{c} \boldsymbol{\mu}_t \\ \boldsymbol{F}\boldsymbol{\mu}_t + \boldsymbol{f} \end{array}\right], \left[\begin{array}{cc} \boldsymbol{\Sigma}_t & \boldsymbol{\Sigma}_t\boldsymbol{F}^T \\ \boldsymbol{F}\boldsymbol{\Sigma}_t & \boldsymbol{F}\boldsymbol{\Sigma}_t\boldsymbol{F}^T + \boldsymbol{\Sigma}_s dt \end{array}\right]\right). \tag{21}$$

The noise covariance $\boldsymbol{\Sigma}_s$ can be obtained by matching both covariance matrices given in Eq. (20) and (21),

$$\boldsymbol{\Sigma}_s dt = \boldsymbol{\Sigma}_{t+dt} - \boldsymbol{F}\boldsymbol{\Sigma}_t\boldsymbol{F}^T = \boldsymbol{\Sigma}_{t+dt} - \boldsymbol{F}\boldsymbol{\Sigma}_t\boldsymbol{\Sigma}_t^{-1}\boldsymbol{\Sigma}_t\boldsymbol{F}^T = \boldsymbol{\Sigma}_{t+dt} - \boldsymbol{C}_t^T\boldsymbol{\Sigma}_t^{-1}\boldsymbol{C}_t \tag{22}$$

The variance $\boldsymbol{\Sigma}_u$ of the control noise is then given by $\boldsymbol{\Sigma}_u = \boldsymbol{B}^\dagger\boldsymbol{\Sigma}_s\boldsymbol{B}^{\dagger T}$. As we can see from Eq. (22) the variance of our stochastic feedback controller does not depend on the controller gains and can be pre-computed before estimating the controller gains.

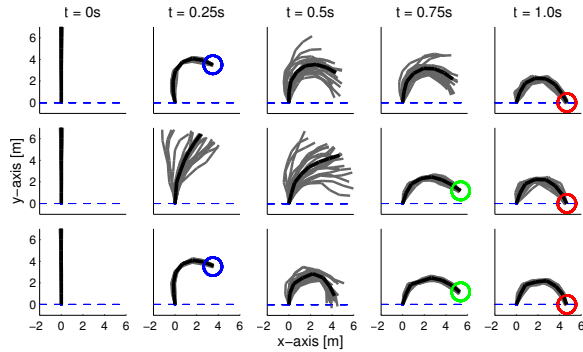

Figure 2: A 7-link planar robot has to reach a target position at $T = 1.0$s with its end-effector while passing a via-point at $t_1 = 0.25$s (top) or $t_2 = 0.75$s (middle). The plot shows the mean posture of the robot at different time steps in black and samples generated by the ProMP in gray. The ProMP approach was able to exactly reproduce the demonstration which have been generated by an optimal control law. The combination of both learned ProMPs is shown in the bottom. The resulting movement reached both via-points with high accuracy.

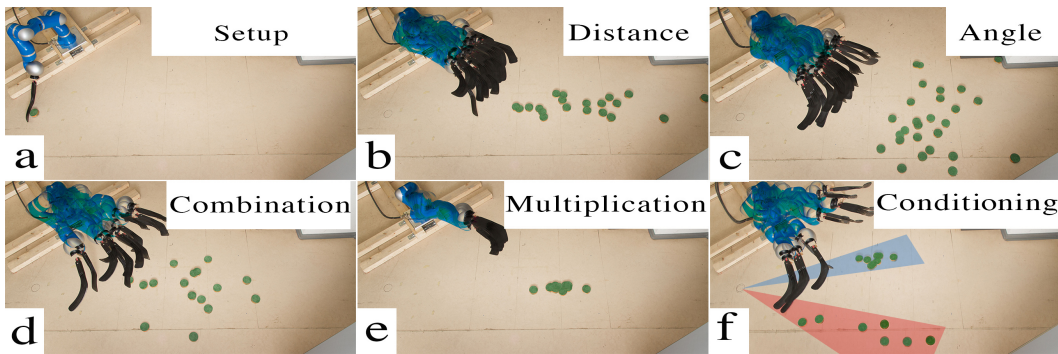

Figure 3: Robot Hockey. The robot shoots a hockey puck. We demonstrate ten straight shots for varying distances and ten shots for varying angles. The pictures show samples from the ProMP model for straight shots (b) and angled shots (c). Learning from combined data set yields a model that represents variance in both, distance and angle (d). Multiplying the individual models leads to a model that only reproduces shots where both models had probability mass, in the center at medium distance (e). The last picture shows the effect of conditioning on only left and right angles (f).

## 3   Experiments

We evaluated our approach on two different real robot tasks, one stroke based movement and one rhythmic movements. Additionally, we illustrate our approach on a 7-link simulated planar robot. For all real robot experiments we use a seven degrees of freedom KUKA lightweight robot arm. A more detailed description of the experiments is given in the supplementary material.

**7-link Reaching Task.**   In this task, a seven link planar robot has to reach a target position in end-effector space. While doing so, it also has to reach a via-point at a certain time point. We generated the demonstrations for learning the MPs with an optimal control law [22]. In the first set of demonstrations, the robot has to reach the via-point at $t_1 = 0.25$s. The reproduced behavior with the ProMPs is illustrated in Figure 2(top). We learned the coupling of all seven joints with one ProMP. The ProMP exactly reproduced the via-points in task space while exhibiting a large variability in between the time points of the via-points. Moreover, the ProMP could also reproduce the coupling of the joints from the optimal control law which can be seen by the small variance of the end-effector in comparison to the rather large variance of the single joints at the via-points. The ProMP could achieve an average cost value of a similar quality as the optimal controller. We also used a second set of demonstrations where the first via-point was located at time step $t_2 = 0.75$, which is illustrated in Figure 2(middle). We combined the ProMPs learned from both demonstrations, which resulted in the movement illustrated in Figure 2(bottom). The combination of both MPs accurately reaches both via-points at $t_1 = 0.25$ and $t_2 = 0.75$.

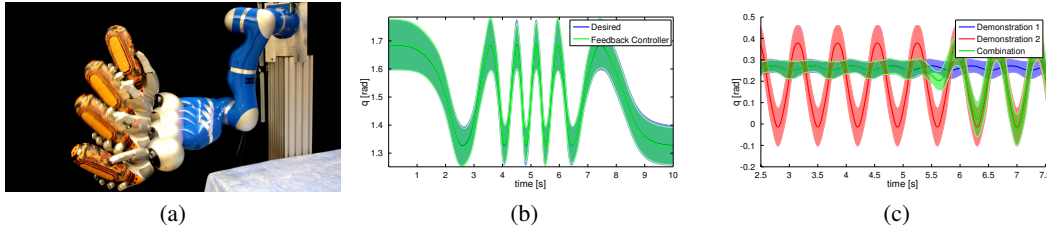

<div align="center">(a)                (b)                (c)</div>

Figure 4: (a)The maracas task. (b) Trajectory distribution for playing maracas (joint number 4). By modulating the speed of the phase signal $z_t$, the speed of the movement can be adapted. The plot shows the desired distribution in blue and the generated distribution from the feedback controller in green. Both distributions match. (c) Blending between two rhythmic movements (blue and red shaded areas) for playing maracas. The green shaded is produced by continuously switching from the blue to the red movement.

**Robot Hockey.**  In the hockey task, the robot has to shoot a hockey puck in different directions and distances. The task setup can be seen in Figure 3(a). We record two different sets of demonstrations, one that contains straight shots with varying distances while the second set contains shots with a varying shooting angle. Both data sets contain ten demonstrations each. Sampling from the two models generated by the different data sets yields shots that exhibit the demonstrated variance in either angle or distance, as shown in Figure 3(b) and 3(c). When combining the two individual primitives, the resulting model shoots only in the center at medium distance, i.e., the intersection of both MPs. We also learn a joint distribution over the final puck position and the weight vectors $w$ and condition on the angle of the shot. The conditioning yields a model that shoots in different directions, depending on the conditioning, see Figure 3(f).

**Robot Maracas.**  A maracas is a musical instrument containing grains, such that shaking it produces sounds. Demonstrating fast movements can be difficult on the robot arm, due to the inertia of the arm. Instead, we demonstrate a slower movement of ten periods to learn the motion.  We use this slow demonstration and change the phase after learning the model to achieve a shaking movement of appropriate speed to generate the desired sound of the instrument. Using a variable phase also allows us to change the speed of the motion during one execution to achieve different sound patterns. We show an example movement of the robot in Figure 4(a). The desired trajectory distribution of the rhythmic movement and the resulting distribution generated from the feedback controller are shown in Figure 4(b). Both distributions match. We also demonstrated a second type of rhythmic shaking movement which we use to continuously blend between both movements to produce different sounds. One such transition between the two ProMPs is shown for one joint in Figure 4(c).

## 4   Conclusion

Probabilistic movement primitives are a promising approach for learning, modulating, and re-using movements in a modular control architecture. To effectively take advantage of such a control architecture, ProMPs support simultaneous activation, match the quality of the encoded behavior from the demonstrations, are able to adapt to different desired target positions, and efficiently learn by imitation. We parametrize the desired trajectory distribution of the primitive by a Hierarchical Bayesian Model with Gaussian distributions. The trajectory distribution can be easily obtained from demonstrations. Our probabilistic formulation allows for new operations for movement primitives, including conditioning and combination of primitives. Future work will focus on using the ProMPs in a modular control architecture and improving upon imitation learning by reinforcement learning.

**Acknowledgements**

The research leading to these results has received funding from the European Community's Framework Programme CoDyCo (FP7-ICT-2011-9 Grant.No.600716), CompLACS (FP7-ICT-2009-6 Grant.No.270327), and GeRT (FP7-ICT-2009-4 Grant.No.248273).

## Footnotes

[1]If inverse dynamics control [20] is used for the robot, the system reduces to a linear system where the terms $\boldsymbol{A}_t$, $\boldsymbol{B}_t$ and $\boldsymbol{c}_t$ are constant in time.

[2]As we multiply the noise by $\boldsymbol{B}\mathrm{dt}$, we need to divide the covariance $\boldsymbol{\Sigma}_u$ of the control noise $\boldsymbol{\epsilon}_u$ by $\mathrm{dt}$ to obtain this desired behavior.

[3]The observation noise is omitted as it represents independent noise which is not used for predicting the next state.

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
