[Supplementary Material · 2013-NIPS-sup.pdf]

# Probabilistic Movement Primitives
# (Supplementary Material)

**Alexandros Paraschos, Christian Daniel, Jan Peters, and Gerhard Neumann**
Intelligent Autonomous Systems, Technische Universität Darmstadt
Hochschulstr. 10, 64289 Darmstadt, Germany
{paraschos,daniel,peters,neumann}@ias.tu-darmstadt.de

## 1    Experiments

We present an analytical description for the experiments conducted in [1].

**7-link Reaching Task.**    We test our approach on a simulated 7-link planar robot. The robot's task is to reach with it's end-effector pre-specified positions on the $x - y$ plane at pre-specified time points. The robot is controlled by setting the desired acceleration $\ddot{q}_{i\in\{1...7\}}$ at each joint and operates in a noisy environment. The environment's noise is additive zero-mean Gaussian control noise with $\mathbf{\Sigma_u} = 200I$. We used two sets of demonstrations, where each set had different via-points and contained $M = 20$ demonstrations. An optimal control planner was used to generate these demonstrations [2]. The optimal control planner used a time-varying quadratic cost function to penalize deviations of the robot's end-effector at the specified via-points. In the first set, we specified via-points at time points $\mathbf{t}^1_{via} = \{0.25, 1\}$s, whereas in the second set we set the via-points at time points $\mathbf{t}^2_{via} = \{0.75, 1\}$s. For training the two ProMPs from these sets, we used $N = 60$ Gaussian basis functions per primitive and dimension. The dimensionality of the weight vector was therefore $\mathbf{w} \in \mathbb{R}^{420}$ and of the covariance matrix $\mathbf{\Sigma_w} \in \mathbb{R}^{420\times420}$. A smaller number of basis functions $N$ would slightly decrease the performance. In the first two rows of Figure 1 we illustrate each of the primitives at different time points, and, in the last row, the combination of the two primitives by multiplying the primitives. The via-point positions are denoted in the figures by the blue, green, and red circles. We observe that ProMPs learned to reproduce exactly the desired distribution in joint space, and, as a result, the robot's end-effector passed over the via-points in task space with a high accuracy. However, during the execution of the remaining movement, the robot exhibited a large degree of variability, matching the demonstrated distribution.

Additionally, we compare on a similar via-point task our approach against the DMPs, and an extension of DMPs as presented in [3]. For the later, we modified our controller such as the proportional part of the gain matrix $\mathbf{K}_t$ to be equal to inverse of the position covariance $\mathbf{\Sigma}_t$, i.e. $\mathbf{K}^P_t = \mathbf{\Sigma}^{-1}_t$, and set the differential gain part $\mathbf{K}^D_t = 2\sqrt{\mathbf{K}^P_t}$ for achieving critical dumping. While the ProMP could achieve an average cost value of a similar quality as the optimal controller, neither the DMP or the extension of DMP could achieve similar performance, as illustrated in Figure 2.

**Robot Hockey.**    In the hockey task, the robot has to shoot a hockey puck to different directions and distances. We used a lightweight KUKA robotic arm with seven DoF, that we control over the Fast-Research Interface (FRI). A hockey stick is mounted as an end-effector. The normal to the stick at the initial position of the robot is used to define the reference frame. The control parameters of the robot at every time point $t_{k\in 1...K}$ are the desired position vector $\mathbf{q}_t \in \mathbb{R}^7$ and the desired acceleration $\ddot{\mathbf{q}}_t \in \mathbb{R}^7$ of each joint. The ProMPs provide at every time point the desired acceleration $\ddot{\mathbf{q}}_t$, while the desired position $\mathbf{q}_t$ is obtained from the second-order Euler integration of the acceleration. The duration of the control step is dt $= 0.001$s. In order to demonstrate the properties of the ProMPs, we used two sets of demonstrations, and we applied different operations between them. The first set contained $M_1 = 10$ demonstrations where the robot shot the puck straight at different distances.

Figure 1: A 7-link planar robot has to reach a target position at $T = 1.0$s with its end-effector while passing a via-point at $t_1 = 0.25$s (top) or $t_2 = 0.75$s (middle). The plot shows the mean posture of the robot at different time steps in black and samples generated by the ProMP in gray. The ProMP approach was able to exactly reproduce the demonstration which have been generated by an optimal control law. The combination of both learned ProMPs is shown in the bottom. The resulting movement reached both via-points with high accuracy.

The demonstrations were provided by a human tutor, using kinesthetic teach-in. The second set also contained $M_2 = 10$ demonstrations where the demonstrator shot the puck at different angles, while trying to keep the variance of the distance relatively small. For each of the training sets, we trained two ProMPs using $N = 10$ Gaussian basis functions per dimensions, which resulted to a weight vector $\boldsymbol{w} \in \mathbb{R}^{70}$. By reproducing the learned primitives, we obtain behaviors illustrated in Figure 3(b) and (c) respectively. We generated the images in Figure 3 by photographing the robot's configuration at the end of the execution of the primitive and when the puck was stopped. The figures show an overlay of the images by multiple executions of each primitive. A new primitive was trained by concatenating the demonstrations of the previous datasets, using the same number of features. The reproduction is depicted in Figure 3(d). We observe that the robot shots with variability on both the angular and distance domain, demonstrating characteristics of both datasets. To demonstrate the probabilistic properties of ProMPs, we generated a new primitive by multiplying the two learned models from the distance and angle demonstrations. The new primitive was generated only by using the trained models from the previous examples, in closed form. Thus, no additional demonstrations or re-estimation of the parameters was needed. The resulting primitive shots the pucks at the intersection of the two models, i.e. straight and at medium distances, as we can see at Figure 3(e). In the last experiment, we selected demonstrations from the angle dataset that either send the puck to the left or to the right and we labeled them accordingly. Then we trained the the parameters of a ProMP by using both left and right demonstrations. In the last step, we trained the joint distribution of the ProMP parameters $\boldsymbol{w}$ and the angle $\alpha$ of the puck at the final position. By conditioning the resulted distribution on the final angle $\alpha$ of the puck, i.e. by computing $p(\boldsymbol{w}|\alpha)$ we obtain a model that shoots in the desired direction, as we can see from Figure 3(f).

**Robot Maracas.** For the rhythmic movement experiment, we used the maracas task. The maracas is a musical instrument containing grains, such that shaking it produces sounds. We used the 7-DoF lightweight KUKA arm with a 15-DoF hand provided by DLR as an end-effector. The hand was used only for holding the maracas and it was not controlled by the ProMPs. Initially, we demonstrate a slow version of the rhythmic shaking movement required to play the maracas. Demonstrating fast movements is difficult due to the inertia of the arm and may also damage the gearboxes of the motors. Hence, a single slow demonstration was provided by the tutor that was about ten periods long. We learned the rhythmic movement using $N = 10$ Von-Misses basis functions per dimension. The ProMP was trained on the $\mathbb{R}^7$ space, including all the DoF of the robot. For facilitating the estimation of the parameters, we split the demonstration in $M = 400$ parts and we assigned the appropriate phase signal. We optimized the parameters of ProMPs using the Expectation Maximization algorithm. After learning the ProMP model for the demonstration, we progressively increase the speed of the movement by modulating the phase, such that the robot successfully plays the instrument. In Figure 4 we illustrate the trajectory distribution of the rhythmic movement while we where modulating the speed of the phase signal at execution time. Different sound patterns where achieved by changing the speed of the movement. Additionally, we demonstrated a second type of movement, which was mainly shaking the maracas sideways. We learned the movement in a similar fashion to the previous one. By defining a smooth blending function, we continuously blend between both movements and produce different sounds patterns. One such transition between the two ProMPs is

Figure 2: Comparison of ProMPs (blue), DMPs (red), and the modified extension of DMPs (green), on the cost function of a via-point task using the 7-link robot. The cost is averaged over $K = 200$ reproductions for every approach. (a) In the first comparison we illustrate the performance of every approach trained with a different number of demonstrations. The DMPs have almost independent performance to the number of demonstrations, since they learn the mean of the demonstrations and they follow the trajectory with constant gains. The extension of DPMs performs better, but it fails to pass though the via-points with high accuracy, and most importantly at the required time step. ProMPs outperform both control approaches. (b) Additionally, we compare the approaches using a different number of basis functions. ProMPs increase their performance slightly, DMPs have already a sufficient amount of basis and they remain at a constant performance level. On the other hand, the ext. DMPs suffer from numerical issues and as a result have a performance impact.

Figure 3: Robot Hockey. The robot shoots a hockey puck. We demonstrate ten straight shots for varying distances and ten shots for varying angles. The pictures show samples from the ProMP model for straight shots (b) and angled shots (c). Learning from combined data set yields a model that represents variance in both, distance and angle (d). Multiplying the individual models leads to a model that only reproduces shots where both models had probability mass, in the center at medium distance (e). The last picture shows the effect of conditioning on only left and right angles (f).

Figure 4: (a)The maracas task. We demonstrate a slow version of the rhythmic shaking movement required to play the maracas. After learning the ProMP model for the demonstration, we progressively increase the speed of the movement by modulating the phase, such that the robot successfully plays the instrument. (b) Trajectory distribution for playing maracas (joint number 4). By modulating the speed of the phase signal $z_t$, the speed of the movement can be adapted. The plot shows the desired distribution in blue and the generated distribution from the feedback controller in green. Both distributions match. (c) Blending between two rhythmic movements (blue and red shaded areas) for playing maracas. The green shaded is produced by continuously switching from the blue to the red movement.

shown for the third joint in Figure 4(c), where in blue and red we show the ProMPs that were learned from the different datasets, and in green the result of the blending.