[Reviews · NeurIPS 2013]

Submitted by Assigned_Reviewer_5

This paper introduces probabilistic movement primitives (ProMP). This is a linear in the parameters, Gaussian, probabilistic representation of movement primitives. The paper has a long section on the advantages of the probabilistic framework, a short section on control and a section on experimental results.

The work is well motivated and there are excellent references to a lot of recent related work.

The model is quite clearly described and in good detail. It is a relatively standard setup for a hierarchical probabilistic model. One issue in the exposition is that it is unclear what the relation ship is between \tau and y? Eg, in eq (1), second equation, the left hand side is a distribution over \tau, but the right hand side is a distribution over y? The derivation of the controller is very compressed, but relatively standard.

The experimental section is very hard to understand. There separate experiments are reported, each illustrating a different capability of the framework. However, the experiment descriptions extend to only about 10 lines each, and it is very difficult to get a feeling for what is going on. For example, I have no idea what the dimensionality of the problems are, and what amount of data is being learnt from and what alternative approaches could possibly have achieved? What is the dimension of the objects inference is made over, eq 5 and 6, how are the priors for eq 5 and 6 chosen, \Sigma_w is potentially a large object (I have no idea of its dimension).

It is a pity that the description of the experiments are 1) too cursory to give much of a feel at all for the method, and 2) no attempt has been made of comparing to any other procedure.

It is not trivial to describe the approach in the very confined space, but important details of the experiment could have been included in supporting material. In the current submission I have difficulty assessing the significance of the paper; with lack of details and lack of comparisons I cannot tell.
Summary: The paper presents a framework for probabilistic movement primitives. The setup of the hierarchical linear in the weights, Gaussian model is fairly standard. There is a good discussion of the benefits of the framework and differences to other existing frameworks, but the experimental section is very limited, contains no comparisons, and lacks that details which would be necessary to evaluate the significance of the approach.

Submitted by Assigned_Reviewer_6

this paper presents a probabilistic framework for using movement primitives for control. the idea is to learn distributions over trajectories by parameterzing as linear functions on some features. parameters represent a single primitive, and these parameter settings can be learned from demonstrations. Given the parameters the primitives are distributions over trajectories so that within the framework probabistic operations can be used to modify or combine primitives e.g. by conditioning on certain points movements can be modified to pass those points, or by multiplying models, one can sometimes perform two desired outcomes simultaneously (focussing on the region where both have mass).

reproducing the trajectories using motor controls is performed using a model based approach, i.e. approximating continuous dynamics with a linear system, linear controller and deriving the control law. One thing that is not clear is how many trajectories do we have to learn controllers for? just the primitives, or once we have a new trajectory (e.g. by blending two primitives) do we have to learn the control law from scratch? Or is are the contols for the primitives combined in some way. This bit of the framework does not seem to be explained.

I think \mu_t needs to be defined as \Psi_t \mu_w before it appears at the beginning of sec 2.3.
Summary: fundamental framework for an interesting framework for control

Submitted by Assigned_Reviewer_7

The paper intends to unify several previously-achieved characteristics of movement primitives (MP) in a single probabilistic framework, while also describing new ways in which this framework allows MPs to be modified or combined. This is accomplished through the representation of trajectories by probability distributions of joint location and velocity. The authors present the foundation for their formulation by describing how standard MP features such as rhythmic and stroke-based movements and temporal modulation are achieved in this new framework. They also discuss how, due to the probabilistic nature of this framework, they can modify position and velocity of a given trajectory through conditioning as well as blend multiple MPs together by multiplying distributions. The authors derive the necessary values for a robotic controller and conclude the paper with experiments on both a real and simulated robotic arm.

The main contribution of this paper is quite obviously the probabilistic characterization of movement primitives. Through the authors' own citations we can see that prior works have already noted the major contributions towards the flexibility and use of MPs in robotic systems. However, as they state, this framework combines several achievements in a single system and provides a new perspective of how to view the theory behind MPs. A problem with their approach is that the increased robustness comes at a cost: The system can no longer be trained with a single demonstration. In their "robot hockey" experiment, the authors state they use a total of 20 demonstrations to train their robot. With such a significant number of demonstrations required, we start to lose some of the benefits of MPs where instead other comparable approaches might be used. Also, in the simulated experiment, the authors state "[t]he ProMP could achieve an average cost value of a similar quality as the optimal controller" yet the variance betwen via-points seems fairly high. I would like to see more discussion as to how optimality is maintained as well as how many samples must be taken from the distribution in order to achieve decent results. A comparison to related Gaussian process models should also be included.
Summary: Interesting application of Gaussian probability distributions used to model variability
of motion primitives.
Author Feedback

Author rebuttal: We thank the authors for their thorough reviews and insightful comments. We apologize for missing explanations and the missing details in the description of the experiments. As the reviewers noted, the shortness of description in some places is due to the page restriction of the format. We will add supplementary material with a more detailed description.

Reviewer 5:
- "One issue in the exposition is that it is unclear what the relation ship is between \tau and y?"
The vector y_t contains the joint positions and velocities at time step t. A trajectory tau is is given by the vectors y_t for all time steps 0 <= t <= T. Eq. 1 defines the probabilistic model for observing a trajectory where we assume there is observation noise on the y_t's. The noise free trajectory is represented compactly
by the parameter vector w. Given w, p(\tau|w) factorizes in its single time steps and is given by the product of p(y_t|\vec w) for all t, as we assumed independent observation noise.

- "The derivation of the controller is very compressed, but relatively standard."
The derivation of the continuous time controller by matching the derivatives of moments is new and can not be found in the literature. We will explain the relation to existing approaches more clearly.

- "However, the experiment descriptions extend to only about 10 lines each, ... \Sigma_w is potentially a large object (I have no idea of its dimension)."
We will add the missing details to the text and also add a supplementary material.
The prior distribution for Equation 5 and 6 is determined by Sigma_w and mu_w. Both define our movement primitive and are learned from multiple demonstrations. In the seven link reaching task the MPs are learned for all seven joints of the robot. We use 60 basis functions per dimension, which results in a w vector with 420 dimensions. In this experiment, we used 20 demonstrations. Fewer demonstrations could be used, however, the quality of the resulting controller in terms of costs would decrease slightly. For both, the hockey and the maracas experiment, the robot also had seven articulated joints that were represented by ProMPs. For each joint we used ten basis functions, i.e. w has 70 dimensions. Ten demonstrations were used for the hockey tasks. For the maracas task we only used 1 demonstration with 10 periods of the movement. This demonstration was split into several data sets
in order to learn a distribution. We used the periodic Von Mises basis functions for the maracas task while for all other tasks, normalized Gaussian basis functions have been used.

- "no attempt has been made of comparing to any other procedure."
We agree that a comparison to other approaches is needed and we will add a comparison in the supplementary material for the seven link reaching task to DMPs, where we use a constant, but optimized feedback controller to follow the trajectory, and the extension of the DMPs given in [12] that can also represent a time-varying variance profile. Our comparison shows that ProMPs outperform both methods in terms of costs of the resulting controller. We will also add an analysis of the resulting costs with an increasing number of demonstrations for all three methods.

Reviewer 6:
- We used 20 demonstrations for the seven link and ten demonstrations for the real robot experiments. When using blending, combination or conditioning, no new demonstrations are needed as the distributions, and therefore the required controllers, can be computed analytically from the original primitive(s).

Reviewer 7:
- The number of required demonstrations is increased due to the increased expressiveness (time varying variance profile, encoding of the covariance of the joints) and the additional benefits (blending, combination, conditioning) of ProMPs. However, if only a small number of demonstrations are given, ProMPs can still be used if we use a proper prior for mu_w and Sigma_w, but we will use some of the beneficial properties. For example, for 1 demonstration, we can use the demonstration as the mean, and a hand-specified diagonal covariance matrix to set the desired variance of the trajectory distribution.

- In the 7-link reaching task, the variance at the via-points is due to the rather high stochasticity in the system. It is in the same range as for the optimal controller (depending on the number of demonstrations, resulting in a 10% to 100% higher costs), while for competing approaches, the costs are up to ten times higher. We agree that this evaluation is very important and we will add it to the paper or the supplementary material.

- "A comparison to related Gaussian process models should also be included."
Most other GP models for control directly learn the policy or a forward model of the system dynamics that is subsequently used for policy search. They do not model the resulting trajectory and hence do not allow for any of the presented operations on the movement representation such as conditioning or blending. Note that the presented model for the trajectory distribution is a special case of a Gaussian process where the prior probabilities for the weight vector w have also been learned, and, hence, a time-varying variance profile can be achieved (see Eq. 4). We will add a comparison to DMPs and the extension of DMPs presented in [12]. ProMPs outperform both methods in terms of costs of the resulting controller.